# Mechanism and Management of Checkpoint Inhibitor-Related Toxicities in Genitourinary Cancers

**DOI:** 10.3390/cancers14102460

**Published:** 2022-05-17

**Authors:** Haoran Li, Kamal K. Sahu, Benjamin L. Maughan

**Affiliations:** Division of Medical Oncology, Huntsman Cancer Institute, University of Utah, Salt Lake City, UT 84108, USA; haoran.li@hci.utah.edu (H.L.); kamal.sahu@hci.utah.edu (K.K.S.)

**Keywords:** genitourinary cancer, immunotherapy, immunotherapy-related toxicity, immune checkpoint inhibitor, immunosuppression, immunotherapy rechallenge, cytotoxic T lymphocyte antigen-4, cytokine

## Abstract

**Simple Summary:**

Immune therapy using checkpoint inhibitors has been increasing in genitourinary cancers for the past decade, starting with monotherapy in renal cell carcinoma and urothelial carcinoma to now include many first-line combinations. More combinations and FDA/EMA indications are undoubtedly on the horizon. An increasing number of patients will be treated with these therapies in the future resulting in a significant increase in the prevalence of immune mediated toxicities. This manuscript focuses on the current understanding of immunotherapy related adverse effects in genitourinary malignancies encountered with the use of immune checkpoint inhibitors.

**Abstract:**

The use of immune checkpoint inhibitors (ICIs) is rapidly increasing as more combinations and clinical indications are approved in the field of genitourinary malignancies. Most immunotherapeutic agents being approved are for the treatment of renal cell carcinoma and bladder cancer, which mainly involve PD-1/PD-L1 and CTLA-4 pathways. There is an ongoing need for recognizing and treating immunotherapy-related autoimmune adverse effects (irAEs). This review aims to critically appraise the recent literature on the mechanism, common patterns, and treatment recommendations of irAEs in genitourinary malignancies. We review the epidemiology of these adverse effects as well as general treatment strategies. The underlying mechanisms will also be discussed. Diagnostic considerations including differential diagnosis are also included in this review.

## 1. Introduction

Genitourinary (GU) malignancies are among the most common types of cancer. According to the World Health Organization, there were over 1.4 million new cases of prostate cancer (PC), 573,278 new cases of urothelial carcinoma (UC), and 431,288 new cases of renal cell carcinoma (RCC) in the year 2020 [1]. Other less common malignancies affecting male organs and the urinary tract include penile cancer, adrenal cancers, and testicular cancer. Adrenocortical carcinoma is a rare GU malignancy with a poor prognosis, with an age-adjusted incident of 1.02 per million population [2].

The development of immunotherapy marks a new era in the treatment of GU cancer. Immunotherapy is the general term for the use of any therapy leveraging the innate immune system or a modified immune system to elicit an anticancer response. The most common immune therapy in genitourinary cancer today includes checkpoint inhibitors. Other immune therapy categories include vaccines, cellular therapies (e.g., bifunctional antibodies or chimeric antigen receptors), and cytokine therapy [3]. Many different co-stimulatory and co-inhibitory checkpoints have been discovered with many therapies currently under investigation directed towards some of these checkpoints [4]. The most common targets in clinical use today include the co-inhibitory targets cytotoxic T-lymphocyte-associated protein 4 (CTLA-4) and programmed cell death protein 1 (PD-1)/programmed death-ligand 1 (PD-L1) [5]. Since 2011, a total of six ICIs being approved by the FDA for GU malignancies, including CTLA-4 inhibitor ipilimumab, PD-1 inhibitors (nivolumab, pembrolizumab), and PD-L1 inhibitors (atezolizumab, avelumab, and durvalumab) [6].

For non-muscle invasive UC, pembrolizumab (200 mg once every 3 weeks or 400 mg once every 6 weeks) is given in high-risk, Bacillus Calmette–Guerin-unresponsive patients. In patients with high-risk UC after undergoing resection, nivolumab is given at 240 mg once every 2 weeks or 480 mg once every 4 weeks. For locally advanced or metastatic UC (mUC), pembrolizumab or atezolizumab (1200 mg once every 3 weeks) can be used in untreated patients who are not eligible for platinum-based chemotherapy [7,8,9,10,11]. Several combination strategies pairing ICIs with additional ICIs or targeted therapies have been also established in treating kidney cancer (pembrolizumab plus axitinib, nivolumab plus cabozantinib, pembrolizumab plus lenvatinib, avelumab plus axitinib, and nivolumab plus ipilimumab) [12,13,14,15,16] after each was proven to be superior to sunitinib. Immune therapy is now the recognized first-line therapy for metastatic RCC resulting in thousands of patients being treated with various combinations for this and other FDA/EMA approved indications.

Our understanding of immunotherapy has been rapidly evolving in the past decade. The toxicity profiles of immunotherapy are distinct from traditional cytotoxic chemotherapy. While they are generally better tolerated, immunotherapies are not without adverse effects. These toxicities are categorically different between the classes of immune therapies: cellular therapy, checkpoint inhibitors and cancer vaccines. No cellular therapies (e.g., CAR-T, BiTE) are yet approved for any GU malignancy. Only one cancer vaccine is approved for GU malignancies, sipuleucel-T a dendritic cell vaccine based on PSA priming [17]. Because of the overall favorable toxicity profile of sipuleucel-T and the unpredictable and clinically challenging toxicities that are present with checkpoint inhibitors, this review will only focus on checkpoint inhibitor related toxicity.

## 2. Epidemiology of irAEs

Immune-related adverse events (irAEs) caused by checkpoint inhibitors share clinical features with autoimmune disease. The most common systems/organs involved in irAEs include skin, gastrointestinal, endocrine, hepatic, renal, and pulmonary organ systems. The severity and organ system affected vary by patient (Table 1). More severe toxicities (CTCAE grade 3–5) are generally infrequent when used as monotherapy and more common when used in combination. In KEYNOTE-045, pembrolizumab for UC, adverse events with any grade occurred in 60.9% of patients treated with pembrolizumab, with 15% of patients discontinuing therapy due to these adverse effects [7]. The most common adverse effects were pruritus (19.5%), fatigue (13.9%), nausea (10.9%), and one pembrolizumab-treated patient died from immune-related pneumonitis [18]. In the JAVELIN Bladder 100 trial, avelumab monotherapy in patients with metastatic UC, the incidence of adverse events from any cause was 98% in the avelumab maintenance group, including 29.4% categorized as irAEs [9]. A total of 7% of patients treated with avelumab developed grade 3 events, but no grade 4 or fatal events occurred. The most frequent irAEs were thyroid disorders (12.2%). In advanced renal cell carcinoma, monotherapy with nivolumab also presented a manageable safety profile in the CheckMate-025 trial [11]. Treatment-related side effects from any cause occurred in 79% of patients treated with nivolumab, including 19% of grade 3 or 4 events. The most common adverse events were fatigue (33%), nausea (14%), and pruritus (14%).

In contrast the combination of a PD-1 inhibitor combined with a CTLA-4 inhibitor is associated with an increased incidence of irAEs [24,25]. In CheckMate-214, nivolumab/ipilimumab for RCC, treatment related adverse events occurred in 93% of participants, including 46% with grade 3 or 4 events. Of those who developed an irAE, 35% received high-dose glucocorticoids. Treatment discontinuation occurred in 22% of patients in the combination group, including eight deaths. The overall safety profile of doublet immunotherapy in CheckMate-214 was more favorable than what was seen in the CheckMate-067, nivolumab/ipilimumab for metastatic melanoma, in which a higher dose of ipilimumab was used demonstrating that CTLA-4 therapy contributes significant added toxicity to this combination [26]. The tyrosine kinase plus PD-1/PD/L1 combination has been shown to also lead to more toxicity than PD-1/PD-L1 monotherapy. The observed adverse events in KEYNOTE-426 which tested pembrolizumab/axitinib in RCC led to treatment interruption in 69.9% of patients and discontinuation in 30.5% of patients [12]. The incidence of adverse events of grade 3 or higher (75.8%) were more frequent than that in pembrolizumab monotherapy from KEYNOTE-427 trial (30.0%), which reported discontinuation because of treatment-related irAEs in 15.5% of patients [27].

One particular challenge with immunotherapy (IO) therapy management is identifying which therapy caused the adverse effect when combination therapy is administered. This is particularly notable as many of the adverse effects can be caused by either agent, such as diarrhea and rashes. However, the management of these symptoms is different for IO-based toxicities compared to tyrosine kinase inhibitor (TKI)-based toxicities emphasizing the need to correctly diagnose the etiology of the symptoms. For example, diarrhea associated with TKI is usually mild and treated with anti-motility agents, while autoimmune colitis caused by IO can be dose-limiting and treatment includes immunosuppression [28]. In JAVELIN Renal 101, axitinib/avelumab for RCC, the most frequent irAEs include hypothyroidism (24.9%) in patients treated with the combination, which can be associated with both agents [15]. The major difference in irAEs between combination and monotherapy is in the frequency of irAEs encountered. In CheckMate-9ER (cabozantinib/nivolumab for RCC) and CLEAR (lenvatinib/pembrolizumab for RCC) we observe that the most common irAEs include dermatitis, thyroid disorders and colitis which is similar what was observed in monotherapy clinical trials [13,14]. However, the frequency of irAE and toxicity overall is much higher than what was observed in Checkmate-025, nivolumab monotherapy in RCC. In CLEAR, dose interruption occurred in 78.4% of patients treated with lenvatinib/pembrolizumab, including 37.2% of patients in this group who discontinued one or both drugs due to adverse effects [14].

The last decade has seen enormous development in the field of GU malignancies with increased FDA/EMA approvals of ICIs in RCC and UC both upfront and as second-line (Table 2). In general, CTLA-4 inhibitors are associated with a higher incidence of irAEs as compared to PD-1/PD-L 1 inhibitors [29]. Combination ICI also have a higher incidence of irAEs [16,30,31,32,33].

## 3. Biochemical Mechanisms for Developing irAEs

The link between PD-1/PD-L1 and autoimmunity has been explored for the past few decades starting with the isolation of PD-1 from a murine T cell hybridoma undergoing programmed cell death [5,37]. In 1999, PD-1 deficiency was identified to result in different autoimmune phenotypes in murine models [38,39]. In these studies, PD-1 knockout mice developed inflammatory arthritis [38]. Nishimura and colleagues found autoimmune dilated cardiomyopathy developed in mice with PD-1 deficiency, concluding that PD-1 is an immune checkpoint contributing to the prevention of autoimmune disease and maintenance of long-term immune tolerance [39]. Unlike PD-1, the role of CTLA-4 in autoimmunity is more complex. While organs infiltrated by lymphocytes were found in mice lacking CTLA-4, the level of autoimmune activity does not quantitatively correlate with cancer immunity [40,41]. In addition, mice with genetic CTLA-4 deficiency die much more rapidly from severe lymphoproliferative disorder [42]. Compared to germline-knockout, adult mice with somatic deletion of CTLA-4 have more indolent disease courses [41]. It is therefore suggested that CTLA-4 acts early in tolerance induction. Each of these checkpoints is also differentially expressed in tissue, with CTLA-4 found primarily in lymph nodes and PD-1 primarily in non-nodal parenchymal tissue. The roles of CTLA-4 and PD-1 appear to regulate immune response differently with CTLA-4 involved at an early phase of T cell activation primarily in lymph nodes, while PD-1 delivers inhibitory signals at a later phase primarily in parenchymal tissues [43,44].

In addition to autoreactive T cells, humoral immunity is also involved in the development of irAEs (Figure 1). In vitro data demonstrate that blockade of the PD-1 pathway increases B-cell activation and production of inflammatory cytokines through the interaction between CD4+ T cells and B cells [45,46]. A subset of CD4+ B helper T cells can stimulate B cell proliferation within inflamed tissues. In PD-1-deficient mice, augmentation of IgG3 antibody response and enhanced proliferation of B cells occurs in a T-cell-independent manner [47]. As a result, some PD-1 knockout mice die from autoantibody-mediated dilated cardiomyopathy [39]. In patients treated with CTLA-4 or PD-1 inhibitors, there is a decline in circulating B cells and an expansion of CD21lo B cells. Those treatment-induced changes correlate with the frequency and timing of irAEs: those with early B cell transition are at a higher risk of developing severe irAEs 6 months after treatment.

Following the activation of immune cells, pro-inflammatory cytokines are released into circulation. Augmentation of these pro-inflammatory cytokines promotes immune cell proliferation, differentiation, and migration [48]. This in turn forms positive and negative feedback loops through multi-level cross-talks between cytokine pathways. For example, production of IL-1α, IL-1β, IL-6, IL-12, TNF-α, and IFN-γ from M1 macrophages drives B cell proliferation and antibody production driving the immune stimulatory and pro-inflammatory phenotype. M2 macrophages release anti-inflammatory cytokines such as TGF-β and IL-10 [49,50,51]. The disruption of immune homeostasis by cytokines appears to play a major role in the development of irAEs [52].

## 4. Predictive Factors for Developing irAEs

Even though our understanding of ICIs has progressed over the past decade, a validated biomarker is lacking to predict which patients will develop irAEs [53]. A reliable biomarker must be clinically easy to use, biologically relevant, and both sensitive and specific. Most of the published literature to date has focused on either tumor factors (such as tumor histology, mutation burden, or genetic variability) or host factors (such as immune cells, cytokines, autoantibodies, body composition, genome/epigenome, or pre-existing conditions) [54,55]. Recent research also suggests extrinsic factors such as the gut microbiome are an integral component of immune homeostasis [56].

Although the precise mechanism remains elusive, tumor histology might have a modestly predictive value on specific irAE profiles. A meta-analysis showed pneumonitis and dyspnea are more likely to occur in mRCC, while arthralgia, hypothyroidism, rash, pruritus, and diarrhea are more likely to occur in melanoma when treated with anti-PD-1 inhibitors [57]. A plausible explanation might lie in tumor-infiltrating immune cells (TIICs). As TIICs are present in both tumor the microenvironment and the affected organs, a different irAE profile may reflect the variation of interaction (immune cell infiltration, neoantigen formation) between TIICs and tumor microenvironment in different cancer types [58].

The ICI or specific combination has a significant influence on the frequency of irAE observed despite the types of irAEs being similar between these classes of ICI. As previously described, PD-1/PD-L1 and CTLA-4 pathways have distinct biological characteristics likely contributing to this difference in frequency. irAEs such as colitis, hypophysitis, and rash were more frequently reported in clinical trials with CTLA-4 inhibitors, whereas pneumonitis, hypothyroidism, arthralgia, and vitiligo were more common with PD-1/PD-L1 inhibitors [57] though all of these complications are observed with each ICI class.

In a study of 137 patients treated with PD-1 inhibitors, preexisting antibodies, such as RF and ANA have been independently associated with irAE incidence as well as clinical benefit [49]. Similar studies on 133 patients treated with CTLA-4 inhibitor showed that newly developed autoantibodies, predominantly anti-TPO (thyroperoxidase) and anti-TG (thyroglobulin), were linked to the occurrence of irAEs and treatment response [50].

Serum cytokines play pivotal roles in the development of irAEs but inconsistently predict for irAEs. Some of the proposed key cytokines and their roles are summarized in Table 3. For instance, increased levels of IL-6 have been associated with IC-induced psoriasiform dermatitis in patients with malignant melanoma (MM) [59,60]. In contrast, patients with MM disease who developed ipilimumab-induced colitis had lower baseline levels of IL-6 [60]. IL-17 is another inflammatory cytokine and is upregulated in patients with inflammatory bowel disease. The baseline IL-17 level was associated with the development of colitis in patients treated with ipilimumab [61,62]. The exact mechanism of the crosstalk between members of the cytokine family and ICI pathway is yet to be demonstrated. Despite this strong mechanistic link, no cytokine has consistently proven to consistently predict irAE risk. Perhaps this is due to challenges in clinical testing related to the rapid degradation of most cytokines in circulation or the rapidly fluctuating concentration of these molecules throughout the inflammatory cycle.

Body mass index (BMI) has emerged as a possible factor in cancer patients receiving PD-1/PD-L1 inhibitors [81]. In an Italian study involving 152 patients with metastatic renal cell carcinoma (mRCC), higher BMI (overweight and obesity) was found to be significantly related to a higher occurrence of any grade irAEs, though only obesity was associated with irAEs above grade 2, as well as the higher occurrence of pulmonary and rheumatic irAEs [82]. In contrast, underlying diabetes, hypertension, hyperlipidemia, the use of metformin or statin were not associated with a higher incidence of irAE.

As only a subset of patients develop irAEs, a genetic link may partially explain this effect. Genome-wide association (GWA) studies have discovered more than 300 susceptibility alleles associated with autoimmune diseases [83]. Though one could argue the pathophysiology of irAE and autoimmune diseases differ in some aspects, growing evidence has shown similar susceptible loci associated with both conditions [84]. Certain human leukocyte antigen (HLA) genes predispose people to developing autoimmune diseases such as myasthenia gravis, myocarditis, hepatitis, and drug rash with eosinophilia and systemic symptoms (DRESS) syndrome [84]. The most convincing evidence exists for loci affecting T-cell receptor signaling pathways such as TNF-α, IL-2, and IL-12 [84,85]. In a retrospective study of patients with metastatic cancer treated with ICIs, Ali et al. suggested a possible association between HLA-DRB1*11:01 and pruritus, as well as HLA-DQB1*03:01 and colitis [86]. Another case–control study found that HLA-DR15, B52, and Cw12 were more prevalent in patients with pituitary irAEs [87].

Germline somatic variation including single-nucleotide polymorphisms (SNP) and polygenetic risk score (PRS) are also associated with autoimmunity. A study of 322 nivolumab-treated patients showed that homozygous PDCD1 804C > T is associated with a decreased risk for irAEs [88]. IMvigor211 is a phase 3 randomized controlled trial comparing atezolizumab monotherapy to chemotherapy in patients with metastatic urothelial carcinoma previously treated with platinum-based chemotherapy [8]. A companion study of IMvigor211 conducted germline whole-genome sequencing (WGS) in 465 individuals enrolled in the trial [89]. Khan et al. found that patients with genetic patterns known to increase the risk of autoimmune diseases such as psoriasis (PSO), vitiligo (VIT), and atopic dermatitis (AD) had an increased risk of skin irAEs during checkpoint blockade. A PRS for PSO, VIT, and AD was predictive of overall survival in IMvigor 211 study [89].

The gut microbiome may also play a key role in maintaining immune homeostasis and therefore irAE risk [90]. Nishio and colleagues investigated ICI responses and gastrointestinal irAEs by transcriptomic profiling gut mucosa from patients who developed immune colitis [91]. The presence of Enterobacteriaceae species increased the likelihood of irAE incidence likely through cytokine-mediated signaling pathways in the colon [91]. Given the similarity between GI irAE and ulcerative colitis (UC), they further compared microbiota and gene expressions in inflamed and uninflamed mucosae from patients with UC and ICI-related colitis [92]. Bacteroides species were decreased in inflamed regions from both irAE colitis and UC supporting the hypothesis that the gut microbiome modulates immune homeostasis and irAE risk [92]. However, the data on the effect of the stool microbiome and irAE risk is complex and somewhat contradictory. It is unknown which species are driving this proposed effect and how meaningful the microbiome is in eliciting irAE host responses.

Ongoing research is identifying many factors that are related to the risk of developing irAE but it appears to be a multifactorial process involving host and tumor factors as well as external influences such as the microbiome. Presently there is no validated tool available for the practicing clinician to determine an individual patient’s irAE risk. Physicians must have a high awareness of these possible complications in order to rapidly diagnose the symptoms and initiate treatment appropriately.

## 5. Treating Patients with Preexisting Autoimmune Disease

Patients with pre-existing autoimmune diseases (AID) have been largely excluded from clinical trials involving ICIs [93]. Many clinicians are hesitant to start ICIs in patients with AID, as clinical trials demonstrate the potential lethal outcomes of severe irAEs from PD-1 and/or CTLA-4 inhibition. In addition, treatment of AID often involves immune suppression therapies, which might compromise the efficacy of ICI. Nevertheless, patients with AID have a wide variation in the severity of their illnesses, thus a personalized approach should be taken in real-world practice [93]. There are limited data on outcomes in this specific population. SAUL is a multinational single-arm safety study of atezolizumab in “real-world” locally advanced or metastatic urinary tract cancers [94]. Out of 997 patients enrolled, 35 patients had AID at baseline, including psoriasis (*n* = 15), thyroid AID (*n* = 6), and rheumatoid arthritis (*n* = 4) [94]. Patients with poorly controlled AID were allowed to enroll. Out of 11 patients who were receiving treatment for active AID, two patients were treated with systemic corticosteroid at the study entry. When treated with atezolizumab, AID patients developed more irAEs of any grade compared to patients without AID (46% vs. 30%) and had more grade 3 to 4 AEs (26% vs. 12%). No significant difference was observed in treatment discontinuation (9% vs. 6%) or treatment-related deaths (0% vs. 1%) between the two cohorts [94]. Interestingly, there is no clear association between baseline AID and type of irAE developed during treatment as some patients developed different irAE relative to the baseline AID and the baseline AID did not always worsen with initiation of atezolizumab. The most common grade 3–4 Aes in AID patients were colitis (*n* = 3), gamma-glutamyl transferase increase (*n* = 2), asthenia (*n* = 2), and hyponatremia (*n* = 2). Among two patients with pre-existing ulcerative colitis, one experienced grade 3 colitis, but the other patient reported no colitis. Most AID flares were managed with steroids (*n* = 11) and methotrexate (*n* = 2). Efficacy of atezolizumab was not inferior in patients with AID despite the prior use of immune suppression compared with the non-AID patients (disease control rate, 51% (95% CI 34–69%) vs. 39% (95% CI: 36–42); median progression-free survival: 4.4 months (95% CI: 2.2–6.3) vs. 2.2 months (95% CI 2.1–2.3). In this first prospective study to include patients with AID, the authors concluded that ICI therapy can be given to patients with pre-existing AID [94,95,96]. One significant limitation of this study is the small number of patients included with pre-existing AID, limiting the confidence in these results.

A systematic review evaluated the retrospective data which included 49 publications, including institutional retrospective studies and case series. Up to 75% (33 to 100%) of patients with pre-existing AID developed irAE, including 41% experiencing an exacerbation of pre-existing AID, 25% with de novo irAEs, and 9% with both [97]. More AID flare-ups were reported with anti-PD-1/PD-L1 agents (62% vs. 36%), while more de novo irAEs were found with anti-CTLA-4 therapy (42% vs. 26%) [97]. Colitis and hypophysitis were the most common de novo irAEs. High-dose corticosteroids were required in 62% of patients with irAEs, and additional immunosuppressive therapies to treat steroid refractory irAEs were given in 16% of patients [16,97].

Most existing data on ICIs in patients with preexisting autoimmune disease originated from melanoma and non-small-cell lung cancer [98]. Data in patients with preexisting AID and concurrently treated with GU cancers is largely unavailable. From the data that is available it appears that patients with AID have similar irAE rates regardless of the underlying malignancy [99].

Because of the high incidence of irAE in patients with pre-existing autoimmune disease, ICIs should only be considered in a selected group of patients. Our recommendation is that patients should meet the following criteria: (1) the preexisting autoimmune disease was mild to moderate in severity; (2) currently the autoimmune disease is quiescent, not requiring active systemic therapy; (3) the autoimmune disease was easily controlled with minimal or no systemic therapy previously. It is recommended that patients be co-managed by a physician specializing in that autoimmune disorder. In contrast, ICIs should be avoided in (1) patients with autoimmune diseases inherently leading to permanent organ dysfunction such as myasthenia gravis or neuromuscular disorders; (2) patients with previous life-threatening or organ compromising disease flares; (3) poorly controlled disease refractory to first-line immune suppressive therapy; (4) active autoimmune disease not currently in remission [100]. This aspect of patient care if rapidly evolving. As more patients are treated our understanding of the risks and benefits will expand. Information learned from the experiences of patients with other diseases will likely improve the care of GU patients with AID treated with ICI [98,99,101,102].

## 6. Management of Immunotherapy-Related Toxicity

### 6.1. General Principles

The incidence of adverse events is highly variable across different clinical trials, with a reported incidence of any-grade toxicity ranging from 15 to 90% [103]. Intensification of treatment is one factor contributing to this variability in reported toxicity. The estimated rate of treatment discontinuation is less when monotherapy is used (13%) compared with combination therapy (43%) [104]. It is highly probably that given the nature of irAEs, and their novelty, the reported rates underestimate the actual incidence [103]. Bertrand et al. studied a total of 1265 patients receiving anti-CTLA-4 antibodies from 22 clinical trials and reported an all-grade irAEs incidence of 72%, and high-grade irAEs observed in 24% of patients [105]. Similarly, in patients receiving anti-PD-1/PD-L1 agents, the reported high-grade irAEs range from 5 to 8% [106]. The incidence increases significantly upon combining the anti-CTLA-4 antibodies and anti-PD-1/PD-L1 agents, with reported high-grade events as high as 55–60% [107,108,109].

A thorough history taking and clinical examination during each clinic visit is essential for early detection to mitigate the risk for morbidity and mortality. The National Cancer Institute/National Institute of Health has defined grading for the adverse events using the standardized Common Terminology Criteria for Adverse Events (CTCAE) grading system [110]. Notably, this system does not separate irAEs from pre-existing autoimmune diseases (Table 4). Key oncology societies have proposed comprehensive guidelines (The American Society of Clinical Oncology, ASCO; The European Society for Medical Oncology, ESMO; and The National Comprehensive Cancer Network, NCCN) to assist oncologists in irAE management [103,111,112].

An appropriate clinical workup is essential to accurately diagnose and grade each irAE (Figure 2). This includes performing necessary laboratory, imaging, and pathology-based studies to assess the extent of organ damage and patient prognosis [113]. For example, in patients who develop acute kidney injury (AKI) after receiving immunotherapy a tissue diagnosis may be helpful in diagnosing ICI related AKI and excluding other causes. However, in patients with a history of nephrectomy, which is often the case in RCC, this might not be practical as a biopsy could present undue harm to the patient in some situations. Therefore, early involvement of organ specialists such as nephrology may be necessary to accurately work through the differential diagnosis [114]. The level of medical care can be either as an outpatient, inpatient (non-ICU), or intensive care depending on the severity of the irAE [115].

### 6.2. Dose Modification of ICIs


Grade 1 toxicity (mild): Usually does not require dose modification. The patient needs close monitoring for any change in the symptoms or worsening of the symptoms.Grade 2 toxicity (moderate): Treatment with ICIs should be temporarily withheld until the toxicity improves to grade 1 or resolves. An exception to this is grade 2 immune-mediated endocrinopathies in which ICI should be held until hormone replacement has been initiated. Patients with immune-mediated endocrinopathies may need a prolonged taper/maintenance of low dose oral steroids (10 mg of prednisone per day or less) only if symptomatic from a treatment flare (e.g., symptomatic hyperthyroidism). Generally, hormone replacement alone is sufficient without concurrent immune suppression for endocrinopathies. Endocrine dysfunction from irAE typically is a permanent complication with life-long hormone replacement needed.Grade 3 and grade 4 toxicity (severe): ICIs are recommended to be permanently discontinued with grade 4; they can be restarted with some grade 3 toxicities after resolution of toxicity to grade 1 or less. High-dose steroids are given with close monitoring for the response. Once toxicity subsides to grade 1 or less, gradual tapering of steroids is recommended over at least 1-month duration.Rapid and aggressive treatment of irAE helps to minimize the incidence of permanent organ injury and severe complications.Corticosteroids are typically the first-line therapy for irAE management.Certain irAEs are treated with upfront combination therapy instead of steroid monotherapy. These typically involve organs associated with a high mortality rate (e.g., cardiomyositis) or organs that easily lead to permanent organ impairment such as ophthalmitis, myasthenia gravis or motor neuropathies.


For additional reading please review [103].

### 6.3. Immunosuppressive Agents to Treat irAEs


Corticosteroids: Steroids remain the backbone of the treatment for irAEs and are usually dosed as 1–2 mg/kg/day of prednisone equivalent. Generally, some clinical improvement is expected within 48–72 h. If no clinical improvement is observed over this duration, then consideration of either intensification of immune suppression or further diagnostic workup for another etiology should be considered. A tailored approach may be required for specific organ involvement with assistance from an organ specialist (Table 4). In the case of steroid-refractory disease alternative immune suppressants are used. Other immune modulators, such as infliximab, mycophenolate, anti-thymocyte globulin (ATG), calcineurin inhibitors, methotrexate, intravenous immunoglobulin (IVIG), and plasmapheresis may be considered on a case-to-case basis [116,117].Infliximab: Infliximab (IFX), a monoclonal antibody functions by binding with high affinity to soluble and transmembrane TNF-alpha which prevents stimulation of TNF-alpha cognate receptors. This reduces pro-inflammatory cytokine levels (IL-1, IL-6). IFX is usually given as a single dose of 5 mg/kg after the failure of oral steroids. Repeat dosing can be administered if needed after 2 weeks of first dose [29,118]. IFX has been found very effective against immune-related colitis and inflammatory arthritis.Vedolizumab: In contrast to the infliximab’s broadly dampening TNFα’s role, vedolizumab (VDZ) is a gut-selective humanized anti-α_4_β_7_ monoclonal antibody [119]. VDZ blocks the interaction between α_4_β_7_-integrin and mucosal addressing cell adhesion molecule 1 (MAdCAM-1) which in turn inhibits the lymphocytic infiltration to the gut [120]. Zou et al. in their observational study compared the efficacy and safety of VDZ and IFX in patients suffering from immune-mediated diarrhea and colitis (IMDC). The study showed encouraging results with fewer hospitalizations (16% vs. 28%, *p* = 0.005), shorter duration of steroid use (35 vs. 50 days, *p* < 0.001), (lower recurrence 14% vs. 29%, *p* = 0.008) while maintaining comparable clinical remission rate (resolution of symptoms to grade 1 or less) in the VDZ arm as compared to the IFX arm (88% vs. 89%, *p* = 0.785). However, the onset of action of VDZ is slower than IFX [17.5 vs. 13 days, *p* = 0.012] [120]. Real-world data also corroborate these findings [119,121].Mycophenolate mofetil: IFX can be hepato-toxic, and hence contraindicated to use in immunotherapy-related hepatitis. In cases of steroid-refractory hepatitis, mycophenolate mofetil (MMF) is typically recommended for steroid-refractory or steroid-dependent drug-induced hepatitis with the usual dose of 500–1000 mg twice daily. MMF is a well-established immunosuppressant used to prevent graft rejection, and autoimmune conditions (autoimmune hepatitis, lupus nephritis, and others) [122,123]. MMF has been successfully used in a variety of irAEs involving nephritis, hepatitis, ophthalmitis, and pancreatitis [124,125,126].Tocilizumab: Tocilizumab is an IL-6 receptor antagonist that has been widely used in treating autoimmune diseases. Retrospective studies suggested that tocilizumab is a viable option for steroid-refractory irAEs [127]. An ongoing prospective trial is assessing the safety and effectiveness in patients who develop steroid refractory irAEs.Intravenous immunoglobulin: IVIG is a pooled IgG derived from healthy donors and is commonly used in a variety of autoimmune conditions. It has a wide range of immune modulation effects on both the B and T cell lymphocyte functions. Its use has been especially explored in neurological irAEs such as Guillain–Barre syndrome (GBS), myasthenia gravis, neuropathies, and hematological irAEs that are thought to be largely antibody mediated [128,129]. There are institutional experiences that have reported better outcomes with upfront IVIG use as first-line therapy when compared to receiving steroids alone [130]. In addition, IVIG has been used to manage autoimmune pneumonitis [131]. A small case series reported seven cases of IVIG-treated steroid-refractory pneumonitis. It suggested a superior efficacy in both oxygenation requirement and mortality, when compared to infliximab treatment [132].Plasmapheresis: The use of plasmapheresis is considered mostly in neurological irAEs, like GBS and myasthenia gravis as second-line therapy in steroid-refractory cases. The reported clinical response is variable in patients with moderate to severe neurological irAEs [116,133,134].


### 6.4. Safety and Risk of Using Immunosuppression

Patients receiving prolonged duration of immunosuppressants are at risk of various secondary complications such as hyperglycemia, hypertriglyceridemia, osteoporosis, and opportunistic infections. Many immunosuppressants have organ-specific complications. For instance, calcineurin inhibitors can cause nephrotoxicity, hypertension, and neurotoxicity [135]. The most common complications with mycophenolate involve the gastrointestinal tract resulting in nausea, vomiting, diarrhea, abdominal cramps, or the bone marrow causing leukopenia and anemia. IVIG and anti-thymocyte globulin can cause infusion reactions, flu-like symptoms, or cytokine release syndrome [136]. The FDA has provided a black box warning for IFX regarding the potential to cause reactivation of tuberculosis and invasive fungal infections [137]. Hence, while it is essential to start immunosuppression immediately for treating irAEs, it is also important to limit the prolonged exposure to immunosuppression. Chronic management of these patients should be conducted under the guidance of a physician experienced with prescribing these medications, such as an organ specialist of an oncologist with significant experience treating steroid refractory irAEs. Currently there are not any prospective clinical trials demonstrating clear benefit of one treatment modality over another in the management of patients with steroid refractory irAEs. The choice of steroid sparing therapy is determined based on clinical factors such as toxicity of the immune suppressive therapy, patient comorbidities and treatment paradigms for related autoimmune disorders (e.g., Crohn’s Disease guidelines for managing patients with immune mediated colitis).

## 7. Impact of Immunosuppression on Survival and Drug Efficacy in Patient’s Treated with ICIs

Patients suffering from metastatic cancer may require steroids for multiple indications such as symptomatic relief of pain, brain edema due to metastasis, fatigue or to manage irAEs. The clinical implications of steroid use on ICIs efficacy and survival outcomes are a matter of ongoing research [138]. There is concern that steroid use or other immune suppressive therapy may attenuate the clinical efficacy of ICI [139]. However, recent studies have shown that steroids when used appropriately do not mitigate the clinical activities of ICI [139,140,141,142]. Horvat et al. showed that OS was not inferior in patients treated with steroids for irAE management compared to patients not receiving steroids (*p* = 0.97) [141]. Petrelli et al. performed a meta-analysis (studies, *n* = 16; patients, *n* = 4045) on patients receiving steroids during ICI for a variety of reasons. Steroid use for any reason in patients treated with ICIs was found to be associated with a 34% higher chance of disease progression or death (HR = 1.34; 95% CI: 1.02–1.76; *p* = 0.03). Similarly, OS was significantly lesser in patients receiving steroids for any reason (HR = 1.54, 95% CI: 1.24–1.91; *p* = 0.0001) than patients without steroids [143]. However, in subgroup analysis, the negative impact on OS was noted in patients who received steroids for supportive care (HR = 2.5, 95% CI 1.41–4.43; *p* < 0.01) or to treat brain metastasis (HR = 1.51, 95% CI 1.22–1.87; *p* < 0.01) but not when used for irAE management (HR = 1.08, 95% CI 0.79–1.49; *p* = 0.62). Similar results were reported by Thompson et al. in their 90-patient study evaluating patients who received high dose steroids for immune-checkpoint inhibitor-related enterocolitis. They found reduced PFS (HR = 2.50, *p* = 0.028) but a similar OS in patients who received steroids as compared to patients who did not receive steroids [144].

In another retrospective study, Arbour et al. reported on the outcome of patients who received steroids at the time of starting anti-PD-L1 therapy [145]. The patients who received baseline corticosteroid use of ≥10 mg of prednisone had decreased progression-free survival (hazard ratio, 1.3; *p* = 0.03) and overall survival (hazard ratio, 1.7; *p* < 0.001). A similar experience was noted by Pan et al., who concluded that patients receiving higher doses of steroids (prednisone > 10 mg/day) for a longer duration (for >2 weeks) have detrimental outcomes during anti-PD1 therapy [138].

Based on these observations, the use of steroids for indications other than treatment of irAE may diminish the efficacy of ICI [138,145]. Repeated studies have shown that higher (>10 mg/day) doses of prednisone for the management of irAEs result in improved clinical outcomes and lower toxicity than lower doses of prednisone (<10 mg/day) [138,142,145]. It is important to treat irAEs aggressively. Care must be taken to avoid under treatment with either low steroid doses, overly short treatment courses, or delayed initiation of therapy all of which are associated with worse organ recovery [142,145,146]. The duration of the steroid dose is guided by the improvement in the irAE, generally with a 2–4-week taper duration recommended [146]. Primary care and ED physicians should be cautious while prescribing steroids for non-cancer-related indications such as nausea. Optimal treatment of other comorbid illness is important to holistically treat the patient. For instance, physicians should prescribe antibiotics, bronchodilators, and steroids for moderate to severe COPD exacerbations in patients actively treated with ICI.

## 8. Issues of Rechallenge with Immunotherapy

For easily managed irAE that have a relatively low risk of causing permanent morbidity/mortality, ICI rechallenge may be safely considered once the steroid is tapered to <10 mg/day prednisone equivalent [147,148,149]. Current oncological guidelines recommend permanent discontinuation of the ICIs for grade 4 irAEs; however, most irAEs are grade 1–2. This means that in most of the non-severe irAEs, rechallenging with immunotherapy is feasible [150,151]. Retrospective studies confirm the feasibility and safety of ICI rechallenge [152,153]. However, close follow-up is needed on rechallenge, because of the higher incidence of irAEs than on the initial exposure. A multicenter study by Ravi et al. evaluated the safety and efficacy of ICI rechallenge in patients with RCC [154]. They reported a 27% of recurrence rate of grade 3 or higher when patients who previously developed grade 3 or higher were re-exposed to ICIs. Development of irAEs with first ICI exposure was associated with a higher risk to develop irAEs on rechallenge. Despite the higher recurrence rate on rechallenge, there were no reported deaths. Similarly, Pollack et al. in their 80-patient dataset reported a recurrence rate of 39% of any irAEs, and 18% incidence of flare of the initial irAE with a median follow up of 14.3 months [151]. Abu-Sbeih et al. found higher recurrence in patients (1) who needed steroids for initial irAEs, (2) required a longer duration immune suppression with the initial irAEs, and (3) with anti-CTLA therapy compared with anti- PD-1/L-1 therapy [155]. Dolladille et al. reported that colitis, pneumonitis, and hepatitis were the irAEs with a higher recurrence rate during rechallenge than adrenal-related irAEs [156]. The likelihood of developing an irAE on rechallenge is higher than in the ICI naïve patient population starting therapy; however, the grade 3–5 incidence appears similar between these two groups from these small datasets reported to date. Importantly the type of organ involvement in the rechallenged patients is unpredictable with a relatively low recurrence of the same irAE which led to prior treatment discontinuation. Haanen et al. reported recurrence of 52% in their 38-patient cohort upon rechallenge, out of which 50% had new irAEs [157]. This underscores the importance of a comprehensive clinical examination and appropriate lab/imaging studies at follow-up to monitor for possible new or different irAEs [157].

ICI rechallenge can be considered after appropriate informed consent with the patient. Factors to consider in optimal patient selection include the risks and benefits of alternative cancer therapy options, previous or ongoing cancer response to ICI in the patient, the severity of initial irAEs, specific organ involvement, ease/difficulty of treatment of initial irAE and pre-existing medical co-morbidities.

## 9. Special Considerations during the COVID-19 Pandemic

The COVID-19 pandemic has broadly impacted oncology practice [158]. Treatment of patients with anticancer agents including immunotherapy during the COVID-19 pandemic creates unique challenges. Overlapping symptoms between COVID-19 and some irAEs such as pneumonitis can make diagnosis more challenging [159]. For instance, symptomatic COVID-19 disease frequently presents with cough, hypoxia, and shortness of breath, which is similar to the presentation of ICI pneumonitis. Similarly, a rise in liver enzymes, diarrhea, elevated troponins, and worsening renal functions are seen both in COVID-19 and ICI-induced hepatitis, nephritis, or cardiomyositis. Early diagnosis and treatment are an essential component of treatment for both COVID illness and irAEs [160,161]. Radiological imaging is nonspecific and is unlikely to be helpful. Testing for SARS-CoV-2 infection can confirm COVID-19 infection and is essential to aiding the diagnosis in these patients. It is reasonable to check for SARS-CoV-2 before initiating immunotherapy or before giving steroids in a suspected case of ICI irAE. Once diagnosed with COVID-19, immunotherapy should be temporarily withheld to allow a complete resolution of COVID-19 [162].

## 10. Conclusions

In this review, we aimed to focus on the development and management of irAEs in GU malignancy, mainly in mUC and mRCC. By better understanding the underlying mechanisms of irAEs, new strategies can be utilized to prevent and treat ICI-related side effects; thus, a better quality of life can be achieved and a broader population can benefit from immunotherapy. During the process of the literature review, however, we found most studies on irAEs were performed in other cancer types such as melanoma and lung cancer. The underlying reason might be that ICIs were first developed and authorized in treating those cancers. Given similarities in mechanisms that are shared across cancer types, there are a limited number of studies from other cancer types being discussed.

Despite the relatively high incidence of irAEs in patients with GU cancers when treated with ICI, most irAEs can be successfully managed with the early initiation of immunosuppressants. Further understanding of mechanism of irAEs will help achieve better treatment outcomes in the future for patients with more severe irAEs

We anticipate that as more immunotherapy drugs are approved in various cancers, the incidence of irAEs is expected to rise. Hence, increased awareness amongst oncologists and primary care physicians is vital to safely prescribe these therapies. Future studies are expected to allow for the development of irAE prediction tools allowing for individual risk stratification and tailoring ICI therapy accordingly.

## Figures and Tables

**Figure 1 cancers-14-02460-f001:**
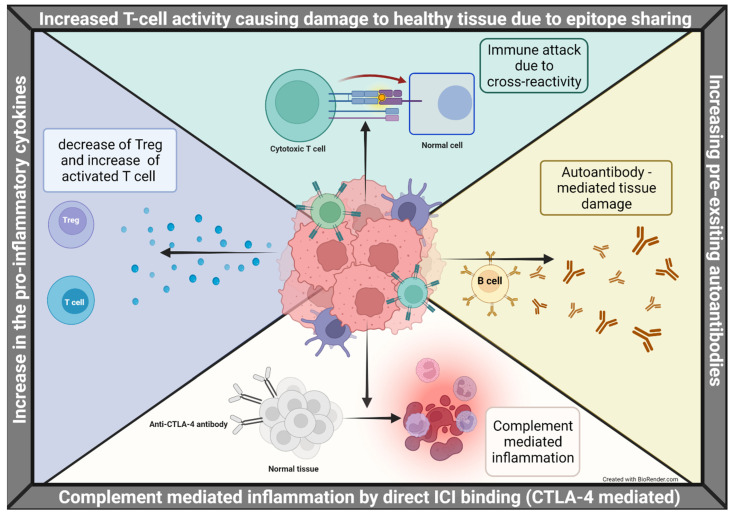
Biochemical mechanisms for developing irAEs. (The figure was created with BioRender.com. Accessed on 11 March 2022).

**Figure 2 cancers-14-02460-f002:**
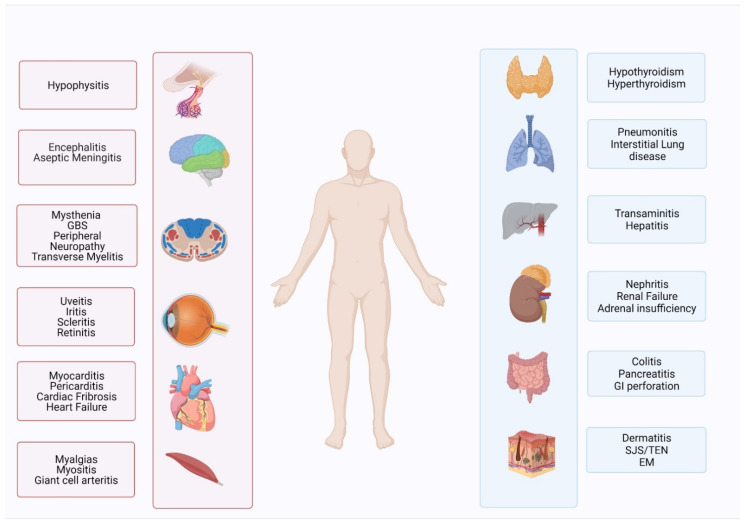
Various organ systems affected by immune-related toxicities.

**Table 1 cancers-14-02460-t001:** Incidence of major irAEs in selected pivotal clinical trials.

Trial Name	AE of Any Grade	AE of Grade 3 or Higher	irAE	Interruption Rate	Discontinuation Rate
KEYNOTE-045 (Pembrolizumab in UC) [19]	62.00%	16.50%	19.50%	Not Available	6.80%
JAVELIN Bladder 100 (Avelumab in UC) [9]	98.00%	47.40%	29.40%	Not Available	11.90%
CheckMate-025 (Nivolumab in RCC) [20]	80.50%	21.40%	Not Available	Not Available	9.60%
CheckMate-214 (Nivolumab/Ipilimumab in RCC) [21]	94.00%	47.30%	85.60%	Not Available	22.10%
KEYNOTE-426 (Pembrolizumab/Axitinib in RCC) [22]	96.30%	67.80%	Not Available	44.00%	12.00%
JAVELIN Renal 101 (Avelumab/Axitinib in RCC) [23]	100.00%	77.88%	45.62%	10.36% (Avelumab only)	23.04% (Avelumab only)
CheckMate-9ER (Nivolumab/Cabozantinib in RCC) [13]	99.70%	75.30%	Not Available	Not Available	19.70%
CLEAR (Pembrolizumab/Lenvatinib in RCC) [14]	99.70%	82.40%	Not Available	78.40%	37.20%

**Table 2 cancers-14-02460-t002:** Key ICI clinical trials in patients with genitourinary cancers.

**Key Studies on Metastatic RCC**			
**Trials**	**Regimens**	**FDA Approval**	**Preferred for**
CheckMate-214 (Motzer 2018) [16]	Nivolumab + Ipilimumab	2018	First-line, for mRCC
JAVELIN Renal 101 (Choueiri 2020, Motzer 2019) [15]	Avelumab + Axitinib	2019	First-line, for mRCC
KEYNOTE-426 (Rini 2019) [12]	Pembrolizumab + Axitinib	2019	First-line, for mRCC
CheckMate 9ER (Choueiri 2021) [13]	Cabozantinib + Nivolumab	2021	First-line, for mRCC
CLEAR (Motzer 2021) [14]	Lenvatinib + Pembrolizumab	2021	First-line, for mRCC
CheckMate-025 (Motzer 2015) [11]	Nivolumab	2015	Second-line, for mccRCC
CheckMate 016 (phase I) (Hammer 2017) [34]	Ipilimumab + Nivolumab	-	-
**Key Studies on Locally Advanced or Metastatic UC**			
**Trials**	**Regimens**	**FDA Approval**	**Preferred for**
IMvigor210 (Rosenberg 2016, Balar 2017) [35]	Atezolizumab	2016	First-line, cisplatin-ineligible
KEYNOTE-045 (Bellmunt 2017) [7]	Pembrolizumab	2017	First-line, cisplatin-ineligible
JAVELIN Solid Tumor Trial (Apolo 2017, Patel 2018) [36]	Avelumab	2017	Second-line, post platinum
CheckMate-032 (Sharma 2016) [33]	Nivolumab	2017	Second-line, post platinum

**Table 3 cancers-14-02460-t003:** Important cytokines that may have an effect on irAEs.

Cytokines	Cytokine Patterns in Patients with Cancer	Main Functions in Cancer	Predictive Value in Immune-Related Adverse Events
Interleukin-2 [63,64,65]	Increased	Immunostimulation	Yes
Interleukin-6 [66]	Increased	Immunostimulation	Yes
Interleukin-10 [67,68]	Increased	Immunosuppression	Yes
Interleukin-12 [69,70]	Decreased	Immunostimulation	Yes
Interleukin-15 [71,72]	Increased	Immunostimulation	Unknown
Interleukin-17 [73,74]	Increased	Immunostimulation	Yes
Interleukin-18 [75,76]	Increased	Immunostimulation	Unknown
Interferon-α [77]	Increased	Immunostimulation	Yes
Interferon-γ [67,78]	Decreased	Immunostimulation	Yes
Tumor necrosis factor α [79]	Increased	Immunostimulation	Yes
Transforming growth factor β [80]	Increased	Immunosuppression	Unknown

**Table 4 cancers-14-02460-t004:** Common terminology criteria for adverse events (CTCAE) and grading system for irAEs, and management with immunosuppression.

CTCAE Grading	Setting of Treatment	Treatment	Immunotherapy
I (Asymptomatic or mild)	Outpatient	Observation	Close monitoring Immunotherapy to continue
II (Moderate)	Outpatient	Low dose steroids (0.5–1 mg/kg/day)	Temporary discontinuation
III (Severe)	Inpatient	High dose steroids (1–2 mg/kg/day)	Consider permanent discontinuation
IV (Life threatening)	Inpatient (likely ICU level)	High dose steroids (1–2 mg/kg/day)	Permanent discontinuation

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
