# Peer review of "Mechanism and Management of Checkpoint Inhibitor-Related Toxicities in Genitourinary Cancers"

_cancers, 2022, doi:10.3390/cancers14102460_

Round 1

Reviewer 1 Report

This review article provided a comprehensive overview on the organ toxicities of immune-checkpoint inhibitors. It briefly reviews the landscape of immunotherapy for genitourinary cancers first and identifies ICIs as the most adverse event-prone of the currently accepted regimens. The remainder of the manuscript expanded to a boarder overview of our current understanding of ICI toxicities.

General comments:

  1. As the topic covers the boarder spectrum on ICI toxicity overall among different cancer types in the later portion of the article, authors may consider editing the title to fit the content better.
  2. The title of the article “recent advances” may be misleading, as the review covers on what has already been found through literature concerning ICI toxicity with some references quoted back in 1999.

Detailed Comments:

  1. Pls review English and grammar across the manuscript as there are several spelling mistakes, suboptimal grammatical choices, and incomplete sentences throughout.
  2. When discussing approved immune checkpoint inhibitors for GU cancers in lines 43-48, please consider including information on how frequently they are used and their efficacy to put into context why this toxicity topic is important.
  3. Line 54-54: “Only one cancer vaccine is approved for GU malignancies, sipuleucel-T a dendritic cell vaccine based on PSA priming”. Pls add reference here.
  4. Line 85-86: “The combination of ICIs with anti-angiogenetic therapy, on the other hand, is not expected to carry synergistic immune-related toxicity.”. This statement may not be precise given the evidence supporting the opposite. Below are the references that demonstrated increased toxicity with the combination.
  • Alessandro Rizzo, Veronica Mollica, Matteo Santoni & Francesco Massari (2021) Risk of selected gastrointestinal toxicities in metastatic renal cell carcinoma patients treated with mmune-TKI combinations: a meta-analysis, Expert Review of Gastroenterology & Hepatology, 15:10, 1225-1232, DOI: 1080/17474124.2021.1948328
  • Fontes-Sousa, M., Magalhães, H., Oliveira, A. et al.Reviewing Treatment Options for Advanced Renal Cell Carcinoma: Is There Still a Place for Tyrosine Kinase Inhibitor (TKI) Monotherapy? Adv Ther 39, 1107–1125 (2022). https://doi.org/10.1007/s12325-021-02007-y
  1. Lines 86-88:” The observed adverse events in KEYNOTE-426 which tested pembrolizumab plus axitinib in RCC led to treatment interruption in 69.9% of patients and discontinuation in 30.5% of patients [12].” It may be more informative to compare these irAEs and treatment interruption/discontinuation rates to pembrolizumab monotherapy? KEYNOTE-427 investigates pembrolizumab monotherapy:
  • McDermott, D., Lee, J.L., Bjarnason, G., et al. Open-Label, Single-Arm Phase II Study of Pembrolizumab Monotherapy as First-Line Therapy in Patients With Advanced Clear Cell Renal Cell Carcinoma. Journal of Clinical Oncology2021 39:9, 1020-1028.
  1. Lines 102-103: “Overall, the safety profile of ICI/TKI combination therapies is consistent and expected compared with monotherapy”. This statement may need further revision to clarify the intention of the authors. In addition, the message regarding the impact on irAE from combination therapy was not consistent throughout the article. Earlier (see comment 3) authors mention that less frequent synergistic toxicity was expected. In line 101 authors mentioned “The difference between combination and monotherapy is in the frequency of irAE encountered.”
  2. Lines 95-97: “Additionally, the management is different for IO therapy-related complications compared, to tyrosine kinase inhibitor (TKI) related complications emphasizing the need to correctly diagnosed the etiology of the symptom despite the inherent challenge.” Maybe worthwhile to add 1-2 lines here to describe the difference in the mechanism of action that contributes to the GI toxicity.
  3. For the section titled Biochemical mechanisms for developing irAEs, some additional potential pathways for irAE development may be considered as below.
  • Ramos-Casals, M., Brahmer, J.R., Callahan, M.K. et al.Immune-related adverse events of checkpoint inhibitors. Nat Rev Dis Primers 6, 38 (2020). https://doi.org/10.1038/s41572-020-0160-6
  • Sophia C Weinmann, David S Pisetsky, Mechanisms of immune-related adverse events during the treatment of cancer with immune checkpoint inhibitors, Rheumatology, Volume 58, Issue Supplement_7, December 2019, Pages vii59–vii67, https://doi.org/10.1093/rheumatology/kez308

  1. Some important and recent literature below may be considered to be added to the discussion in the Predictive biomarkers for developing irAEs section:
  • Jia, XH., Geng, LY., Jiang, PP. et al.The biomarkers related to immune related adverse events caused by immune checkpoint inhibitors. J Exp Clin Cancer Res 39, 284 (2020). https://doi.org/10.1186/s13046-020-01749-x
  • Hommes, J., Verheijden, R., Suijkerbuijk, K., Hamann, D. Biomarkers of Checkpoint Inhibitor Induced Immune-Related Adverse Events—A Comprehensive Review. Oncol. 2021. https://doi.org/10.3389/fonc.2020.585311
  1.  In the section on Treating patients with pre-existing autoimmune disease, some relevant literature below may be useful to add to the discussion.
  • Boland P, Pavlick AC, Weber J, et al. Immunotherapy to treat malignancy in patients with pre-existing autoimmunity. Journal for ImmunoTherapy of Cancer 2020;8:e000356. doi: 10.1136/jitc-2019-000356
  • Coureau M, Meert AP, Berghmans T, Grigoriu B. Efficacy and Toxicity of Immune -Checkpoint Inhibitors in Patients With Preexisting Autoimmune Disorders. Front Med (Lausanne). 2020;7:137. Published 2020 May 7. doi:10.3389/fmed.2020.00137
  • Kehl KL, Yang S, Awad MM, Palmer N, Kohane IS, Schrag D. Pre-existing autoimmune disease and the risk of immune-related adverse events among patients receiving checkpoint inhibitors for cancer. Cancer Immunol Immunother. 2019;68(6):917-926. doi:10.1007/s00262-019-02321-z
  • Gong, Z., Wang, Y. Immune Checkpoint Inhibitor–Mediated Diarrhea and Colitis: A Clinical Review. JCO Oncology Practice 2020 16:8, 453-461

  1. Lines 384-390: IVIG is also reported for managing pulmonary irAEs. See the below references:
  • Balaji A, Hsu M, Lin CT, et al Steroid-refractory PD-(L)1 pneumonitis: incidence, clinical features, treatment, and outcomes. Journal for ImmunoTherapy of Cancer 2021;9:e001731. doi: 10.1136/jitc-2020-001731
  • Trinh S, Le A, Gowani S, La-Beck NM. Management of Immune-Related Adverse Events Associated with Immune Checkpoint Inhibitor Therapy: a Minireview of Current Clinical Guidelines. Asia Pac J Oncol Nurs. 2019;6(2):154-160. doi:10.4103/apjon.apjon_3_19

Reviewer 2 Report

In this review, Li et al describe the use of immune checkpoint inhibitors in patients with GU malignancies, and also the management of irAEs associated with ICIs. While this is an important topic, I do have some comments outlined below, which limit my enthusiasm a bit. The article in general is a bit difficult to read given the lack of flow, grammatical issues, lack of headers, lengthy paragraphs, and lack of figures.

Major:

-Some of the writing could be clearer, and there are grammatical errors throughout. I would recommend thorough re-reading and editing by a native English speaker

-The overall objective of this review is not clear, and there is discordance between the title, the abstract, and the content of the article. The title suggests the article will focus on ICI-related toxicities, but then the abstract highlights the use of CAR-specific therapies. Would ensure that there is a clear focus/message for the review – is it immunotherapy in general? Or just ICIs?

-The introduction is far too long, and it’s difficult for the reader to digest. If the focus is only on checkpoint inhibitor toxicity as the authors claim on page 2, line 58, then would significantly pare down the preceding discussion

-Review articles are much easier to digest if there are headings throughout, shorter paragraphs, and several figures. Would strongly suggest making these changes. For instance, instead of presenting a series of percentages on page 2, lines 59-73, why not make a bar graph to show the percentages of immune-related adverse events across trials? Also- in relation to the above comment about the intro being rather long, would perhaps create a separate section related to the “Epidemiology of irAEs”

-Figures are easier to digest than lengthy paragraphs. Under “biochemical mechanisms for irAEs,” would make a representative figure showing the mechanism/pathways for irAEs

-The authors provide “strong” recommendations for the treatment of patients with autoimmune disease, but the authors stated earlier that there are limited data in this population – would be cautious about saying that these are “strong” recommendations when data in this area are limited (page 6, lines 276-288)

-Management of irAEs: there is little discussion about the workup of irAEs, which is highly variable depending on the organ system involved. The discussion is also very general, rather than specific to patients with GU malignancies. As described earlier in the review, patients with GU malignancies often receive combination therapies which make the diagnosis difficult. In our subspecialty, we think a tissue diagnosis is critical for diagnosing ICPi-AKI, but this can be difficult in patients with a history of nephrectomy, as is often the case in patients with a history of RCC. Some of these management issues as they pertain to patients with GU malignancies should be elaborated on further

Minor:

-As noted above, several grammatical and spelling errors throughout. Page 2, line 45 “paring” should be “pairing”

-Table 1: would consider rearranging these trials such that they are in chronologic order

-Page 5, line 192 – BMI is not a “biomarker.” This is a risk factor. Would separate out risk factors from biomarkers. Similarly, paragraph 2 under “predictive biomarkers” (page 4, lines 161-169), describes mechanisms and not biomarkers, and likely belongs in the preceding section

-It is not clear how lines 176-180 on page 4 fit in with the rest of the paragraph- the authors talk about the combination of ICIs and toxicity profiles, and then segue to talking about how pre-existing antibodies are associated with irAEs. These are 2 separate issues, and therefore should be in separate paragraphs-Immunosuppressive agents, pages 8-9: what about IL-6 inhibitors like tocilizumab?

Reviewer 3 Report

The work is potentially interesting
corrections are needed to improve quality

1. There are some errors in the references such as the erroneously cited reference No. 5 regarding the autonomy line 115, and I think some other reference needs to be put.

Part 2 of the text binds cytokines and ineffective responses should be stitched (line 182). It is known that in the serum of patients with tumors there are many cytokines that are elevated in the advanced stage of the disease and they may be the reason for ineffective therapy as previously shown in the papers:
Multiomic analysis of cytokines in immuno-oncology. Expert Rev Proteomics. 2020 Sep; 17 (9): 663-674.
Cytokine patterns in patients with cancer: a systematic review.
Lancet Oncol. 2013 May; 14 (6): e218-28.
3. Among these negative cytokines, in addition to IL-6, TNF-beta and TGF-betta are certainly very important.
4. It has been shown that TNF-α leads to induction of apoptosis and cleavage of molecules from the membrane by proteolytic action and thus performs immunomodulation and reduces the effectiveness of therapy, which should certainly be added and discussed. TNF-alpha induced apoptosis is accompanied by rapid CD30 and slower CD45 shedding from K-562 cells. J Membr Biol. 2011 Feb; 239 (3): 115-22.
5. In general, the whole paper is general, and more examples should be given for urinary tumors, but the authors describe the whole text in general for any tumor without paying attention to the genitourinary tract and its specifics.

Round 2

Reviewer 1 Report

Thank you for addressing the comments. I have no further questions.

Author Response

Thanks for the help in reviewing this manuscript!

Reviewer 2 Report

I appreciate the responses to my comments.

Author Response

(The authors gave the same response as above.)

Reviewer 3 Report

the authors did not respond adequately and did not correct the paper completely

Author Response

For point 1, we have sent this manuscript for native speaker to edit prior to resubmission.

For point 2-4, we have added a new table (Table 3) to address the function and possible predictive value of cytokines in the development of irAEs, including TNF-alpha and TGF-beta. 

For point 5, we have added a comment on the conclusion section for explanation, 

"In this review, we aimed to focus on the development and management of irAEs in GU malignancy, mainly in mUC and mRCC. By better understanding the underlying mechanisms of irAEs, new strategies can be utilized to prevent and treat ICI-related side effects, thus a better quality of life can be achieved and a broader population can benefit from immunotherapy. During the process of literature search, however, we found most studies on irAEs were done in other cancer types such as melanoma and lung cancer. The underlying reason might be that ICIs were first developed and authorized in treating those cancers. Given similarities in mechanisms that are shared across cancer types, there are a limited number of studies from other cancer types being discussed."

Thanks again for help reviewing this manuscript